# Disorder-dominated quantum criticality in moiré bilayers

Yuting Tan [1] ✉, Pak Ki Henry Tsang[1] & Vladimir Dobrosavljević [1] ✉

Moiré bilayer materials have recently attracted much attention following the discovery of various correlated insulating states at specific band fillings. Here we discuss the metal-insulator transitions (MITs) that have been observed in the same devices, but at fillings far from the strongly correlated regime dominated by Mott-like physics, displaying many similarities to other examples of disorder-dominated MITs. We propose a minimal theoretical model describing the interplay of interactions and disorder, which is able to capture all the universal aspects of quantum criticality, as observed in experiments performed on several devices.

The field of the metal–insulator transitions[1,2], which still retains an aura of mystery and mystique[3], is living a veritable revolution. The principal obstacle, from the experimental perspective, is the challenge to carefully tune to the transition point, while avoiding the effects of spurious charge, spin, or orbital orders, which can mask the genuine mechanisms associated with the MIT[3]. This difficult quest has suddenly shifted in high gear over the last few years. An extraordinary flurry of activity was triggered by the recent discovery of moiré bilayer materials of various kinds, which allows unprecedented control over the physical properties of the electron systems at hand. Narrow bands have been engineered[4], which can be carefully tuned both in terms of the bandwidth and the band filling, allowing precise and systematic studies of several regimes of interest around various insulating states. A number of correlated insulators have indeed been discovered[5] at partial band fillings, signaling the dominance of electron–electron interactions in the narrow band limit. While the intricate interplay of electron correlations and band topology[6] remains a fascinating subject of ongoing debate for moiré graphene bilayers, a somewhat simpler situation is found in moiré transition metal dichalcogenide (TMD) bilayers. Here, genuine Mott–Hubbard physics was theoretically predicted[7] and observed[8] close to half filling ($f = 1$, one electron per moiré cell).

A remarkable recent paper[9] documented such an approach to the Mott point, by electric field control of the bandwidth at half filling. The reported transport behavior, as well as thermodynamic response, displayed all the characteristic features previously established in other Mott systems, such as the "spin-liquid" molecular materials[10,11]. The same study, on the other hand, demonstrated very different behavior in a regime far away from half-filling, where strong correlation effects

should not play a significant role. Here, the magnetic field response has indeed proven to be remarkably mild, suggesting the lack of spin localization, which is the hallmark of Mott physics. Nevertheless, a MIT was clearly observed upon bandwidth tuning at integer band filling $f = 2$ (two electrons per moiré cell), which so far has not been a subject of much scrutiny. A closer look at the experimental data reveals several interesting signatures, which clearly distinguish this regime from the behavior around the Mott point.

The following features stand out (see Fig. 1a), data provided by the experimental group[12], and Supplementary Fig. 6[9]): (i) On the metallic side, the resistivity displays linear-$T$ behavior at low temperatures: $R(T) \approx R_0 + AT$, with $A > 0$ further away from the transition. This result is in dramatic contrast to what is found in the same device around the Mott point ($f = 1$), where the Fermi Liquid $T^2$ law is very clearly seen[9]. Its absence here hints at the lack of strong correlation effects away from half-filling. (ii) As the transition is approached, the slope $A$ initially increases, reaches a maximum, and then decreases again towards the transition. An (almost) "flat" curve, sometimes called the "separatrix"[13] (or the "Mooij point"[14]) is seen before the transition is reached. This behavior is similar to what is frequently observed in disorder-driven MITs[13,15]. (iii) A characteristic temperature scale $T^*$ marks the extent of the leading linear-$T$ regime, and is seen to decrease towards the transition. (iv) The residual conductivity $\sigma_0 = 1/R_0$ also decreases steadily, extrapolating to zero past the "Mooij point"—exactly as in many other examples of disorder-driven MITs[3]. (v) The activation gap $\Delta$ displays a similar decrease on the insulating side, interpolating to zero at precisely the same point where conduction vanishes, suggesting a continuous transition. We follow the same procedure used in ref. 9 to

[1]Department of Physics and National High Magnetic Field Laboratory, Florida State University, Tallahassee, FL 32306, USA. ✉e-mail: ytan@magnet.fsu.edu; vlad@magnet.fsu.edu

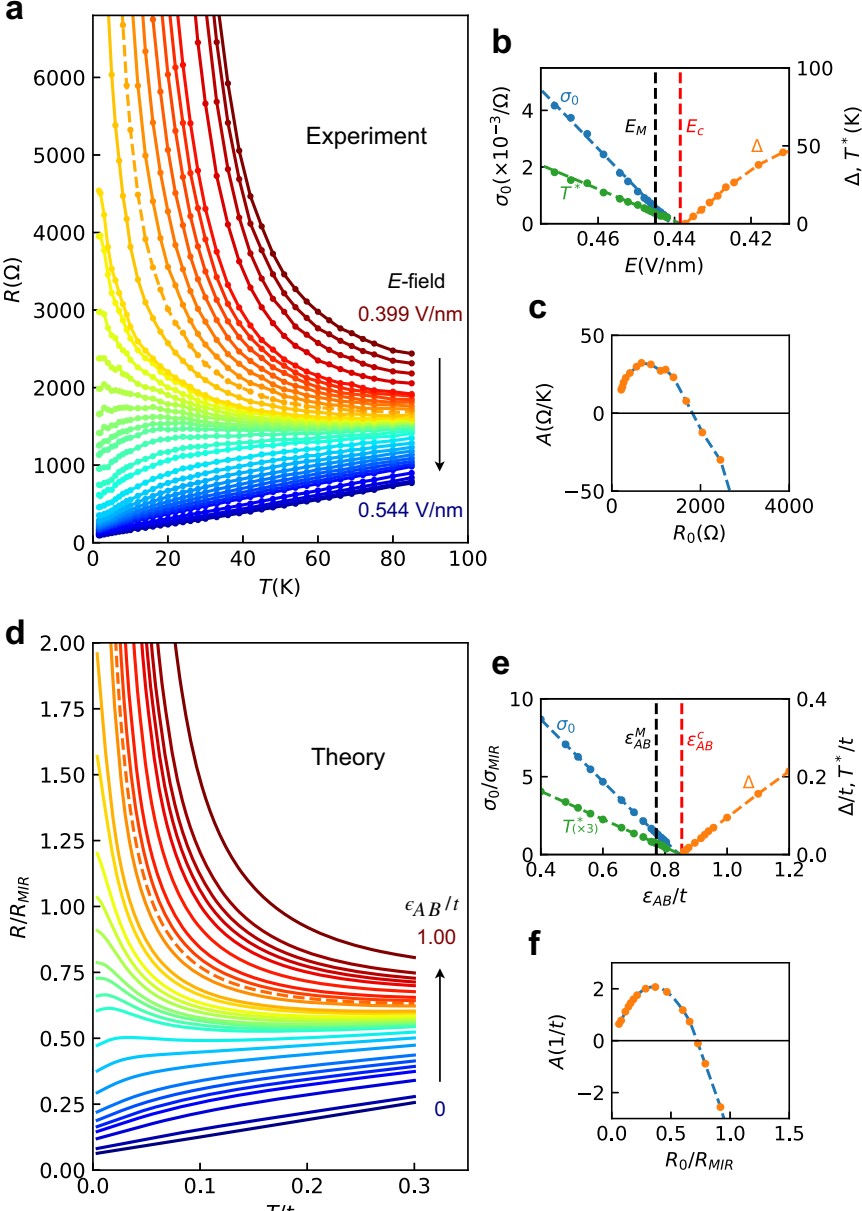

**Fig. 1 | Transport behavior across the MIT at integer band filling.** All the qualitative features found in experiments (upper panel) are captured by our CPA-DMFT theory (bottom panel). **a** Experimental $R(T)$ curves provided by experimental group[12], with 0.399 V/nm $\leq E \leq$ 0.544 V/nm; **d** theoretical curves for $0 \leq \epsilon_{AB}/t \leq 1.00$. The dashed lines in **a**, **d** are critical curves $R_c$. The insets **b**, **e**, show the extrapolated $T = 0$ conductivity $\sigma_0$, the boundary of linear resistivity region $T^*$, and the activation energy $\Delta$, as function of electric field $E$ (experiments) or band separation $\epsilon_{AB}$ (theory). $E_M$, $E_c$ correspond to experimental Mooij point and critical point, while $\epsilon_{AB}^M$, $\epsilon_{AB}^c$ were obtained from theory. **c**, **f**, The slope $A$ describing low-$T$ resistivity $R(T) \approx R_0 + AT$, displays non-monotonic behavior as a function of $R_0$. The same analyses are also performed on data in ref. 9 (Supplementary Fig. 6).

extract the activation gap. The details of how to extract $A$, $T^*$ are given in Supplementary Note 3 (Supplementary Fig. 5).

In the rest of this paper, we present a robust physical picture that is able to capture all the qualitative (and even some quantitative) trends seen in the experiment. At filling $f = 2$, the narrow moiré band is completely full, but the corresponding band gap shrinks as the bands broaden, eventually leading to band overlap and metallic behavior[9]. When band overlap is modest, one expects electron and hole pockets with very small Fermi surfaces for charge carriers, which in this regime become vulnerable to even modest amounts of disorder. At finite temperatures, charge transport is also affected by additional scattering from thermal excitations, which is further enhanced in the dilute carrier limit. Thermal and impurity effects, however, cannot be clearly

decoupled in this regime of poor conduction, as generally found also for many other disorder-driven metal–insulator transitions[16]. This leads to a nontrivial interplay of interactions and disorder, and the associated change of sign of $A = dR(T)/dT$, the "Temperature Coefficient of Resistivity" (TCR) preceding the MIT, a widely observed phenomenon sometimes called the "Mooij correlation"[14]. All these features can be captured in a self-consistent theory of interactions and disorder[14,16], which can be viewed as the minimal model for disorder-dominated MITs in (moderately) interacting electron systems. It describes how certain interaction effects are generally enhanced in presence of disorder, leading to strong disorder renormalization, which in some cases also triggers polaron formation. This physical picture differs significantly[3] from (non-interacting) Anderson

localization, illustrating the seminal ideas of Phil Anderson himself going back to 1970s[17]. It also predicts the precise form of the scaling behavior for the family of resistivity curves (see below), thus formulating a concrete phenomenology that can be very useful in analyzing future generations of experiments.

## Results

### Model of interactions and disorder

Motivated by the experimental setup in moiré TMD bilayers[9], we consider a two-band model of electrons at integer band filling, in presence of moderate disorder, and where interaction effects are represented by the coupling of carriers to a bosonic field[16]. Here we do not specify the physical origin of the bosonic excitations, which could be soft phonons specific to the bilayer structure[18], but could also represent the response of other low-energy collective modes[19–21] to single-particle displacements. Guided by experiments, which clearly demonstrate the absence of Mott-like physics at integer filling, we ignore the spin degree of freedom and thus any significant role of the on-site Hubbard $U$, or the possibility of any kind of magnetic order. Our model is described by the following Hamiltonian:

$$
\mathcal{H} = -t \sum_{\langle i,j \rangle} c_i^\dagger c_j + \sum_{i \in A} \epsilon_A c_i^\dagger c_i + \sum_{i \in B} \epsilon_B c_i^\dagger c_i \\
+ \sum_i (\xi_i - \mu) c_i^\dagger c_i + g \sum_i X_i (c_i^\dagger c_i - n) + H_b,
$$

(1)

where $c_i^\dagger (c_i)$ are the creation (annihilation) operators for spinless electrons hopping between sites $i$ and $j$ of a triangular lattice, with hopping integrals $t$. The two-band model (Fig. 2a, Supplementary Fig. 1) is obtained by periodically modulating the site energies within a unit cell consisting of three sites, with one site in the unit cell (sublattice B) having site energy $\epsilon_B$, while the other two sites (corresponding to the two degenerate sublattices) have energies $\epsilon_A$. We define the "band splitting" energy $\epsilon_{AB} = \epsilon_A - \epsilon_B$. Extrinsic disorder is characterized by a random distribution of site energies $\xi_i$, with a uniform distribution of the form $P_o(\xi) = \frac{1}{W} \theta((\frac{W}{2})^2 - \xi^2)$, where $W$ measures the disorder strength. In addition, the electrons interact locally with dispersionless bosons of frequency $\omega_o = \sqrt{K/M}$, described by $H_b = \sum_i KX_i^2/2 + P_i^2/2M$. We use $t = 1$ as our unit of energy. The strength of electron-boson coupling is measured by the dimensionless coupling constant $\lambda = \frac{g^2}{2KD}$, where $2D = 9t$ is the bare bandwidth of our triangular lattice. In addition, the lattice filling $n = \frac{1}{N} \langle \sum_i c_i^\dagger c_i \rangle$ is kept constant at $n = 1/3$, giving a band insulator (lowest band fully occupied) in the split-band limit.

## Comparison of theoretical and experimental features

In the following, we present a detailed solution of our model, which due to its simplicity can be analytically solved in several limits, while the corresponding numerical solution can be obtained with any desired accuracy. To be specific, we select the following values of the model parameters $g = 1$, $K = 1$, and $W = 1$, corresponding to moderate disorder ($W/2D \approx 0.1$) and moderate electron-boson coupling ($\lambda \approx 0.1$). The evolution of the single-particle density of states (DOS) $\rho(\omega)$ at $T = 0$, as a function of band splitting at $\epsilon_{AB}$ is shown in Fig. 2b. For $\epsilon_{AB} = 0$, it resembles the conventional density of states of the triangular lattice, however with some rounding introduced by disorder. When $\epsilon_{AB}$ increases, the DOS at the Fermi energy starts to decrease, until a hard insulating gap forms at $\epsilon_{AB}^c = 0.854$, indicating the MIT. Near the critical point, $\rho \sim \omega^{1/3}$, as clearly shown with larger $W$ and broader critical region (see Supplementary Figs. 3 and 4). The corresponding residual conductivity $\sigma_0$ (blue dashed line in Fig. 1e) decreases linearly as the MIT is approached: $\sigma_0 = \sigma(T = 0) = 1/R(T = 0) \sim \delta^\mu$, where $\delta = (\epsilon_{AB}^c - \epsilon_{AB})/\epsilon_{AB}^c$ measures the distance to the transition and the conductivity exponent is $\mu = 1$. Our model can be further solved at finite temperature, producing the entire family of resistivity curves (Fig. 1d), similar as in the experiments, which we now analyze in detail.

As in the experiment, the theoretical curves exhibit linear-T behavior at low temperatures on the metallic side of the transition. The evolution of the slope (TCR) with external field exactly matches the experimentally-observed trends, as can be seen from (Fig. 1c and f). The slope $A$ (and $R_0$) initially increases upon application of the electric field, because the number of available carriers decreases as the size of the electron (hole) Fermi pockets shrink (Supplementary Fig. 2). At larger fields, the trend reverses, recovering the "Mooij correlation" behavior expected when disorder becomes dominant. This phenomenon, which implies the breakdown of Matthiessen's rule, generally precedes MIT itself[14], and is caused by the buildup of correlations between the increasingly inhomogeneous electronic density and the bosons responsible for thermal scattering.

An additional energy scale characterizing the metallic regime is $T^*$, the boundary of the linear-T region, which decreases linearly towards the transition as $T^* \sim \delta$. Similar behavior is also found for the activation gap $\Delta \sim \delta$, which describes the approach to the transition from the insulating side. Remarkably, all the qualitative trends and the values of the critical exponent $\mu = 1$ predicted by our model precisely match the experimental findings.

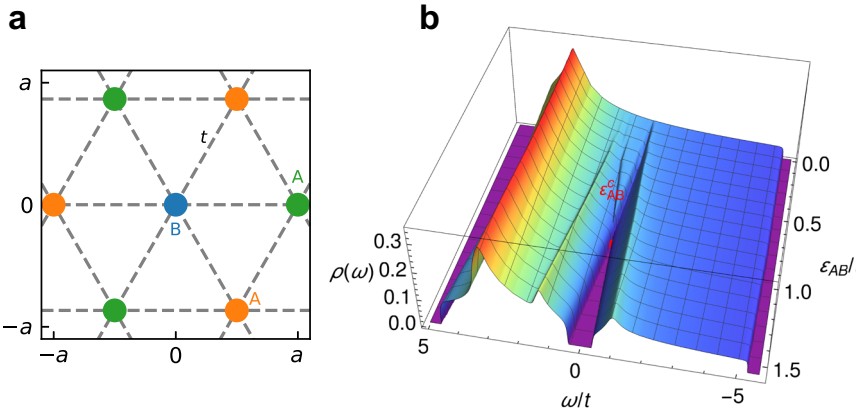

**Fig. 2 | Moiré lattice and the density of states. a** Triangular lattice with lattice spacing $a$, hopping parameter $t$, and three sites in unit cell. One site has site energy $\epsilon_B$ (blue dot), while the other two have site energy $\epsilon_A$ (orange and green dots). **b** Disorder-averaged single-particle density of states per site $\rho(\omega)$ for different band separation $\epsilon_{AB}$; here $\omega$ is measured with respect to the Fermi energy. The bands split beyond critical separation $\epsilon_{AB} > \epsilon_{AB}^c = 0.854$, producing a continuous metal–insulator transition, where all quantities display power-law behavior.

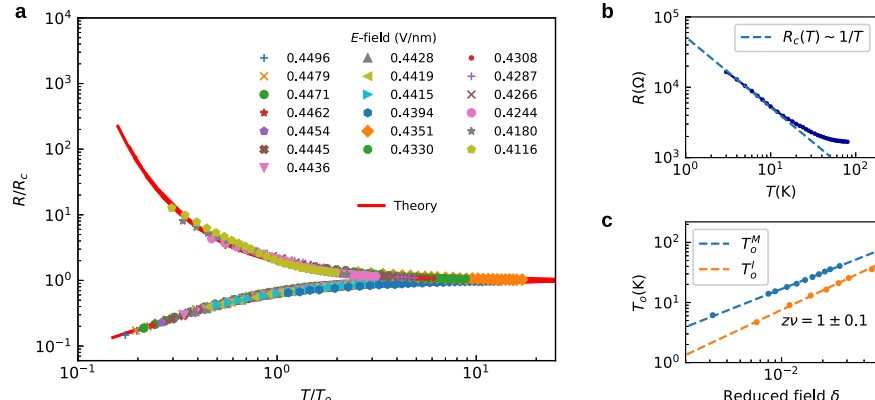

**Fig. 3 | Scaling analysis for both experimental and theoretical results. a** Scaling analysis is performed for the experimental resistance curves corresponding to Fig. 1a, within 0.412 V/nm ≤ $E$ ≤ 0.450 V/nm, revealing near-perfect agreement with the theoretical scaling function (red line), with no adjustable parameters. **b** Critical resistance $R_c \sim T^{-x}$ on experiments displays behavior consistent with the theoretical prediction for exponent $x = 1$ (Supplementary Fig. 9). **c** Crossover temperature $T_o(\delta)$ obtained from the scaling collapse of experimental data (the superscripts $M$ and $I$ designate respectively the metallic and insulating branch. The estimated critical exponent $z\nu \approx 1 \pm 0.1$ is consistent with the theoretical value $z\nu = 1$.

## Quantum critical scaling

A direct analysis of the resistivity curves (Fig. 1; see insets), both experimental and theoretical, clearly indicates a continuous (i.e., quantum critical[22]) character of the MIT. On very general grounds, the presence of criticality implies scaling behavior of various observables, and in the specific case of the MITs, we expect[3] the resistivity $R(T, \delta)$ to take the form

$$R(T, \delta) = R_c(T) f(T / T_o(\delta)). \qquad (2)$$

Here $R_c(T) = R(T, \delta = 0) \sim T^{-x}$ is the critical resistivity curve, $f(y)$ is a universal scaling function, and $T_o \sim \delta^{\nu z}$ is the crossover temperature associated with the approach to quantum criticality. Guided by these expectations, we next perform the appropriate scaling analysis[3] to both the experimental (range: 0.412 V/nm ≤ $E$ ≤ 0.450 V/nm; dots in Fig. 3a) and the theoretical resistance curves (range: $0.802 \leq \epsilon_{AB} \leq 0.905$).

We identify the value of the critical field in the experiments, at which a simple power-law dependence of $R_c(T)$ is observed (Fig. 3b), and we find the exponent $x \approx 1$, consistent with our theory (Supplementary Fig. 9). Remarkably, this power-law behavior occurs precisely at the same critical field where $\sigma_0$, $T^*$, and $\Delta$ all extrapolate to zero (see Fig. 1), further confirming the quantum-critical character of our transition, both in the experiment and in theory. We then normalize $R(T, \delta)$ by the critical resistance $R_c(T)$, and after rescaling $T$ by a field-dependent factor $T_o(\delta)$, the curves collapse onto two branches, as shown in Fig. 3a (dots). We emphasize that in implementing such an "unbiased" scaling procedure[3], we do not assume any specific form for the field dependence of the crossover temperature $T_o(\delta)$. Instead, we directly verify that it indeed vanishes at the critical point, by plotting it as a function of $\delta$ on a log-log scale, as shown in Fig. 3c, giving the experimental estimate for the critical exponent $z\nu = 1 \pm 0.1$ (here the error estimate reflects the uncertainty associated with the scaling collapse procedure for the experimental data).

The scaling of the theoretical resistance curves is performed following an identical procedure, with $T_o(\delta)$ shown in Fig. 4 (red dashed line), giving the theoretical exponent $z\nu = 1$. This analysis (Supplementary Note 5) also provides us with the precise quantitative form of the universal scaling function $f(y)$ for our model (red line in Fig. 3a), which we can directly compare to the experimental findings without any adjustable parameters. A similar analysis is also performed for another experimental data set[9] (Supplementary Figs. 6, 7, and 8), providing further support for our picture. Although we have used a relatively broad range of fields and temperatures in analyzing the experimental

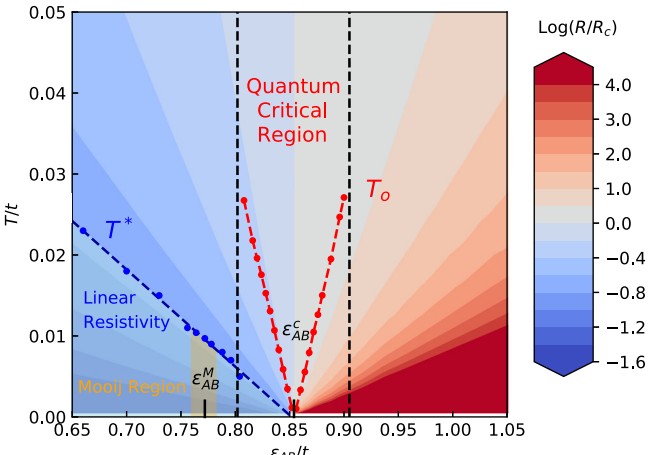

**Fig. 4 | Theory phase diagram for disorder-driven MIT.** Blue dots $T^*$ (dashed line obtained by fitting), serving as the boundary of the linear-T resistivity region, extrapolate linearly to $\epsilon_{AB}^c$. The so-called Mooij region $0.760 < \epsilon_{AB} < 0.782$ (yellow), where the slope $A \sim R_0$, is sitting around the Mooij point $\epsilon_{AB}^M = 0.772$. Red dots are the theory scaling temperatures $T_o$, indicating the critical region $0.802 < \epsilon_{AB} < 0.905$. The critical exponent $z\nu = 1$. The scaling argument shows that the critical region can not go past the Mooij point. The respective color codes of Log($R/R_c$) are given to the right, which displays a fan-shaped pattern.

data, we find a remarkable quality of scaling. Almost-perfect agreement with the theoretical prediction is found, not only concerning the estimated values for all critical exponents but also for the precise form of the scaling function $f(y)$. We also observe that, both in the experiment and in theory, the metallic and insulating branches are quite asymmetric with respect to each other. This is precisely what one generally expects for disorder-dominated transitions, where the resistivity displays only modest temperature dependence on the metallic side, while it is generally exponentially strong in any insulator. We should mention that such behavior is in dramatic contrast (Supplementary Note 7) to what is seen for the Mott transition ($f = 1$ curves experimentally obtained[9] for the same device), which reveals pronounced "mirror symmetry" of the scaling function[23], consistent with both microscopic theory[24] and careful experiments[25,26] on other Mott systems.

We should emphasize that, on general grounds, a universal scaling behavior should be expected[27] only in the immediate vicinity of the

critical point, and not necessarily over an extended range across the phase diagram. A natural question, therefore, is how large is the critical region in the case we consider here, especially concerning the non-monotonic behavior and the vanishing of the TCR parameter $A$ in the Mooij correlation regime[14] (Fig. 1). In the following, we use general scaling arguments to demonstrate that strictly speaking the critical region cannot intercept with the Mooij regime, as shown in the theoretical phase diagram in Fig. 4. To start with, the critical resistance has a simple power-law dependence on temperature: $R_c \sim T^{-1}$. It is obvious that, on the metallic side, the slope $A = \partial R / \partial T \to -\infty$ as we approach the transition. On the other hand, the condition of having finite resistivity $R_0$ in the $T \to 0$ limit (within the metallic phase) requires that, for $y = T/T_0(\delta) \ll 1$, the scaling function must assume the form $f(y) = y^\alpha(1 + ay^\beta + \cdots)$. It follows that $R(T, \delta) \sim T_0^{-\alpha}T^{\alpha-1}(1 + a(T/T_0)^\beta + ...)$, and $a$ is a universal constant. Since $R(T, \delta) \approx R_0 + AT$ on the metallic side, we conclude that $\alpha = \beta = 1$, and to leading order $A \sim T_0^{-2} \sim \delta^{-2}$. This means that $|A|$ decreases monotonically away from the transition, but it cannot change the sign within the critical region, where leading power-law scaling is obeyed. Indeed, since $T_0 \sim \delta^{\nu z} \sim \xi^{-z}$ (where $\xi$ is the relevant correlation length associated with the critical point), $|A|$ becoming small indicates that the corresponding correlation length also becomes short, marking the boundary of the critical region. All these features are very clearly seen in examining the details of our theoretical solution (Supplementary Fig. 10). On the other hand, the relevant violations of scaling, as introduced by the change of sign of $A$, prove to remain parametrically small in a very broad range of parameters, much beyond the Mooij point. In this interval, approximate scaling behavior is observed, with the same scaling function describing the strict critical regime. Exactly the same situation is found in experiments, where the theoretical scaling function collapses the experimental data within a surprisingly broad range of parameters, displaying quantitative agreement with theory without any adjustable parameters.

## Discussion

We presented and solved a minimal theoretical model for disorder-dominated transitions, as motivated by the experiments on moiré TMD bilayer materials at integer band filling. It paints a physical picture of bosonic modes which strongly renormalize the potential energy landscape seen by the mobile electrons at the verge of band splitting— driving a continuous metal–insulator transition. We showed that this transition displays all the features generally expected for a disorder-dominated MIT, revealing critical behavior in a striking agreement between theory and experiments. Ours is the first example of a theory for disorder-dominated MITs that is able to fully explain all the universal aspects of a real experimental system, which represents a significant step forward for this age-old problem residing at the heart of solid-state physics.

## Methods

### Theoretical

In our model, we neglect the on-site Hubbard U as well as the long-range Coulomb interaction. We do so because we focus on the $f = 2$ regime (proximity to the band insulator), which is very far from half-filling ($f = 1$). Here the onsite Coulomb repulsion U is generally not expected to play any significant role, nor is the long-range component of the Coulomb interaction. The latter is expected to play a dominant role in the formation of Wigner crystals at low band filling ($f \ll 1$)[28,29]; in our case, the tendency to form such charge-ordered states is suppressed due to reduced compressibility. To solve our model we use a self-consistent theory of interactions and disorder[14], which combines Dynamical Mean Field Theory for the interaction effects and the Coherent Potential Approximation for describing the effects of disorder. Similarly, as for the popular SYK model[30], this theory becomes an exact solution both in the limit of infinite range hopping or for large coordination. Details of the calculations can be found in Supplementary Note 1, where we also

show how to use the Kubo formula to calculate the corresponding transport properties within this approach.

Because we attribute the linear-$T$ behavior of the resistivity to incoherent electron-boson scattering above an appropriate Debye scale, which can be very low in energy[18,31], we can ignore the dynamics of the bosons, which in turn enables a fully self-consistent solution of the problem in the semi-classical (thermal) regime. For the same reason, the actual form of the boson dispersion is irrelevant to our purposes and we ignore it. As a matter of fact, a close look at the experimental data (Fig. 1a) reveals that the resistivity deviates from linear behavior at the very lowest temperatures ($T < 1$ K), which can be viewed as the lower boundary for the validity of our semi-classical treatment. Describing the interplay of thermal bosonic excitations with disorder within a poor metal is the central goal of our theory. This mechanism should not be confused with "Strange Metal" behavior[30] found in many Mott materials and other examples of strongly correlated matter. The latter is not likely to be of relevance in the regime around integer band filling we consider, where the strong correlation effects are neither expected nor experimentally detected[9].

### Experimental

The additional experimental data (Device 2) we analyze in this paper were obtained in the same fashion as the data presented in ref. 9, at $f = 2$ (Device 1), but on a different device.

## Data availability

The experimental data that support the findings of this study are provided in Source Data files. The theoretical data generated during this study are available from the corresponding author upon reasonable request. Source data are provided with this paper.

## Code availability

The custom codes generated during this study are available from the corresponding author upon reasonable request.

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

## Acknowledgements
We thank Jie Shan, Kin Fai Mak, and Tingxin Li for very useful discussions and for kindly sharing with us some unpublished experimental data (Device 2). We are also grateful to Simone Fratini, Sergio Ciuchi, Oskar Vafek, and Louk Rademaker for useful comments. This work was supported by the NSF Grant No. 1822258 (V.D.), and the National High Magnetic Field Laboratory through the NSF Cooperative Agreement No. 1157490 and the State of Florida (V.D.).

## Author contributions
Y.T. carried out theoretical calculations and perform the analyses. P.K.H.T. provided technical support and helped do the calculations. V.D. designed the project. Y.T. and V.D. discussed the data, interpreted the results, and wrote the paper.

## Competing interests
The authors declare no competing interests.
