## [Peer Review File · Nature Communications]

REVIEWER COMMENTS

Reviewer #1 (Remarks to the Author):

The bandwidth-tuned metal-insulator transition (MIT) in moiré TMDs at filling $f=2$ shows interesting phenomenology that has not received as much attention to date as the corresponding MIT at $f=1$. In this work the authors present a theoretical model which describes the $f=2$ transition, including the effects of relevant interactions and disorder. The model developed by the authors does well in describing many of the experimental observations, such as the non-monotonic behavior of the TCR and the critical scaling of resistivity curves, providing evidence for the mechanism driving the $f=2$ MIT seen pioneering in experiments by the Cornell group.

The results seem sound and valid and the manuscript is well written. I believe this study will be of interest in the community of moiré materials, motivating further experiments at $f=2$. For that reason I would recommend the manuscript for publication after the following points/questions are addressed:

1. The bosonic excitations play a crucial role in the mechanism for the transition. The T-linear behavior of the resistivity in the metallic phase at low temperatures is attributed to scattering between electrons and the bosonic modes below a Debye scale. If the bosonic modes were for instance acoustic phonons as in reference 18 for TBG, what would be the order of their characteristic energy/temperature scale? Does it coincide with the temperature at which the resistivity starts to deviate from linear in the experiments?
2. Besides on-site interactions, long-range and non-local interactions have been shown to be relevant in moiré systems. Will they play a role here? Is there an argument to neglect them?
3. Would the mechanism driving the transition also apply in situations when the spin-valley degree of freedom of TMDs is included?
4. Regarding CPA, why does this local approximation work so well in reproducing the experimental results for this particular system in contrast to others?

5. Disorder can have different origins, such as charge impurities or twist angle disorder (although for aligned bilayers this is expected to be small). Can all sources of disorder be treated on the same footing by the distribution used here? Interactions introduce a self-energy that renormalizes the disorder, how is the disorder distribution renormalized as δ changes?

-- As a side comment, maybe the same kind of experiments on heterobilayers rotated at a small twist angle could reach the strongly disordered regime (larger values of λ) where the criticality is expected to be different. --

6. What is the scaling for the calculated DOS? In some parts of the text it says $\rho \sim \omega^{1/3}$ and in some others $\rho \sim \omega^3$.

7. By looking at the very good agreement between theory and experiments, most probably it is not necessary to go beyond mean-field theory to describe the general trends of this transition. However, can the authors discuss in which limits/conditions should their approximation break down?

8. In the data from device 1 in Fig. S6, A does not become negative, is this attributed to a less disordered sample? Are estimates of the disorder amounts within each device available?

9. I recommend adding the reference Phys. Rev. Lett. 121, 026402 next to reference 7 in the introduction, this was the theoretical prediction of Hubbard model physics in moiré TMDs. Reference 7 was the experimental confirmation.

10. In the caption of Fig. 1, could the authors mention what the vertical black and red lines stand for in panels B and E. It is said in the main text but not on the figure.

11. In the caption of Fig. S6, "three characteristic properties A, T and Δ ..." should be σ_0 instead of A.

12. After Eq. (S5) it reads "hybridazation" instead of hybridization. After Eq. (S10) "bonson" probability distribution.

In their manuscript “Disorder-dominated quantum criticality in moiré bilayers” the authors Yuting Tan, Pak Ki Henry Tsang, and Vladimir Dobrosavljević propose and investigate a model for quantum criticality in moiré bilayers of transition metal dichalcogenides (TMDCs). In contrast to the strongly correlated regime, where the systems is dominated by the Mott-Hubbard physics, at the doping levels of interest in this work, the metal-to-insulator transition (MIT) is found to be disorder-driven.

The results of the manuscript are interesting to a broad readership of condensed matter physicists and represent an important step towards the understanding the physics of moiré TMDCs. The text is clearly written and the scientific analysis of the model seems to be sound. I therefore recommend the publication in Nature Communication if the following points are properly addressed:

Main points

1. The central point of the paper is the explainability of quantum critical behavior at integer filling of moiré TMDCs by disorder effects. In the manuscript I miss a discussion of what is actually known about disorder in the (experimentally investigated) TMDC devices. This could include, for instance, the type of the disorder and the disorder strength. Is the uniform disorder proposed by the authors applicable to the manufactured devices from experimental evidences or is it justified rather a-posteriori?
2. In the introduction the authors discuss nicely the differences of their modelization of quantum criticality via disorder with the Mott criticality. The later was recently investigated in detail within a dynamical mean-field theory study of the so-called moiré Hubbard model in a preprint of Jiawei Zang, Jie Wang, Jennifer Cano, Antoine Georges, and Andrew J. Millis (arXiv:2112.03080). The authors should contrast their results to the ones obtained in this preprint. Especially, there, the authors also demonstrate scaling behavior in both the metallic and insulating part of the phase diagram. Please also comment on the “mirror symmetry” in this respect.
3. In Fig. 4 the authors show a nice phase diagram as a function of temperature and band-splitting. I was wondering how this phase diagram would be changing if the disorder strength and/or the electron-boson coupling were changed? Is the existence of the Mooji region bound to a specific choice of the (relative) values of these parameters? In this respect I was a bit confused by the notion of “comparison to experimental findings without any adjustable parameters” (l. 156-157). The authors should clarify this statement.

Minor points (and typos, etc.)

4. What is the filling f referring to? Is this the filling with or without displacement field?
5. Is there any sign of a magnetic ordering tendency appearing in the CPA-DMFT calculations?

6. Typos

- l. 12: “on in” → “on”.
- l. 26: “TMD” → “transition metal dichalcogenide (TMD)”.
- l. 35: “here indeed proved” → “has indeed proven”.
- l. 127: “the the” → “the”.
- l. 221: “examples strongly” → “examples of strongly”.
- l. S11: “bosons) for triangular” → “bosons) for the triangular”.
- l. S19: “hybridazation” → “bosons) for the hybridization”.
- l. S25: “green” → “Green”.
- l. S74: Rc → R_c .
- l. S76: “give” → “gives”.
- l. S88: “mention” → “mentioned”.
- Fig. S2: “Color” → “Colors”.

Response to Reviewers' comments (NCOMMS-22-16629-T)

We sincerely thank both Reviewers for taking the time to review our work carefully and prepare very perceptive reports. To summarize, both Reviewers found our results solid and sound, and interesting to a broad readership of condensed matter physicists, especially in the community of moiré materials, but asked for clarification on several points concerning our model and the methods we used. In the following, we provided a point-by-point response to the Reviewers' comments. We have revised the manuscript and the supplementary based on the Reviewers' suggestions accordingly (marked red as required), and we are confident to have addressed points/questions to their satisfaction. Please find the list of main changes at the end of this reply. We believe the Reviewers' insightful comments have led to significant improvements in the readability and completeness of our manuscript.

Reviewer #1 (Remarks to the Author):

R1.1

The bandwidth-tuned metal-insulator transition (MIT) in moiré TMDs at filling $f=2$ shows interesting phenomenology that has not received as much attention to date as the corresponding MIT at $f=1$. In this work the authors present a theoretical model which describes the $f=2$ transition, including the effects of relevant interactions and disorder. The model developed by the authors does well in describing many of the experimental observations, such as the non-monotonic behavior of the TCR and the critical scaling of resistivity curves, providing evidence for the mechanism driving the $f=2$ MIT seen pioneering in experiments by the Cornell group.

The results seem sound and valid and the manuscript is well written. I believe this study will be of interest in the community of moiré materials, motivating further experiments at $f=2$. For that reason I would recommend the manuscript for publication after the following points/questions are addressed:

Response R1.1

We are glad that the Reviewer finds our results sound and interesting. We also thank the Reviewer for recommending publication, and for the very helpful suggestions, which led to significant improvements in our manuscript.

R1.2

1. The bosonic excitations play a crucial role in the mechanism for the transition. The T-linear behavior of the resistivity in the metallic phase at low temperatures is attributed to scattering between electrons and the bosonic modes below a Debye scale. If the bosonic modes were for instance acoustic phonons as in reference 18 for TBG, what would be the order of their

characteristic energy/temperature scale? Does it coincide with the temperature at which the resistivity starts to deviate from linear in the experiments?

Response R1.2

While direct numerical comparison of band-structure and phonon parameters for TBG and TMD hetero-bilayers can only be suggestive at the order-of-magnitude level, it is indeed interesting to comment on the relevant phonon energy scale in the case of TBG. In this case, estimates in Ref. 19 suggest that the relevant energy scale above one expects for linear-T resistivity is of the order of 5-10 K. This result is generally consistent with the experimental transport results for TBG, although the sole relevance of phonons for the observed linear-T resistivity still remains under debate. The precise phonon energy scales for various TMD hetero-bilayers are still poorly understood at present, since the field is under very quick development. However, one does generally expect the presence of relatively low-energy phonons with the relevant energy scales in the range of several K (J. Shan and F. Mak, private communications).

In the case of TMD hetero-bilayers studied experimentally in Ref. 9, one observes linear-T resistivity over a broad range of temperatures, $T_{\min} < T < T^*$, where T_{\min} is generally in the range of only a few K, roughly consistent with the TBG behavior and theory. These results strongly suggest the validity of the model presented in our paper, for the purpose of elucidating the transport trends around the $f = 2$ metal-insulator transition in a broader T-range of relevance to these experiments.

We should stress that, as a matter of principle, our model does not necessarily rely on the existence of phonons as a source of thermal scattering. Other low-energy bosons could also be considered, as we mention in the manuscript, but a more detailed investigation of the microscopic models for such bosons is beyond the scope of our paper. Still, several recent theoretical studies have given examples of such low-energy bosons in various moiré materials, which could in principle be of relevance. In addition to the already cited work of Crépel et al. (Ref. 20), in the revised version of the manuscript we also added a reference to the work of Vafeek and Kang (Ref. 21), which predicts the emergence of certain low-energy Goldstone modes in the context of TBG.

R1.3

2. Besides on-site interactions, long-range and non-local interactions have been shown to be relevant in moiré systems. Will they play a role here? Is there an argument to neglect them?

Response R1.3

The on-site interaction U is neglected in our model mainly because we focus on the $f = 2$ case (proximity to a band insulator), which is very far from the half-filling, where onsite repulsion between spin up and spin down electrons is generally expected not to play a significant role. Indeed, the characteristic magnetic field dependence, which was beautifully documented in Ref. 9 in the Mott ($f = 1$) regime, was reported to be almost entirely absent at $f = 2$ we considered

here. For the same reason, we expect the non-local spin interactions to also be relatively unimportant in this regime.

The long-range Coulomb interaction, in contrast, normally plays a dominant role if one considers the formation of various Wigner crystals at low band filling ($f \ll 1$) (For example, Ref. 28 and Ref. 29). In such situations, where charge ordering breaks translational invariance and causes insulation, the long-range Coulomb interaction takes central stage. Such Wigner-Mott states have indeed been observed at $f \ll 1$ in the new TMD hetero-bilayer moiré systems, but they have not been observed around $f = 2$. On general grounds, if there exist other reasons for insulation, such as the proximity to a band insulator or a Mott insulator, there arises an electronic gap that is about to open, which suppresses charge fluctuations. In such situations, the non-uniform electronic states are very costly in energy so the long-range Coulomb interaction is usually not as important.

Given the above general arguments, we expect that our minimal model for the considered $f = 2$ situation should be able to capture the main trends, as we indeed demonstrated, even without including explicitly the Coulomb interaction. We thank the Reviewer for this important question. To clarify the matter, we now added an appropriate discussion to the manuscript (Line 207 - 216).

R1.4

3. *Would the mechanism driving the transition also apply in situations when the spin-valley degree of freedom of TMDs is included?*

Response R1.4

In TBG, non-local magnetic inter-site interactions such as direct-exchange are important because of the effects of (“weak”) topology. That effect is probably what contributes to the insulating states in TBG. In TMD homo-bilayers (*Nature* **597**, 345–349 (2021)), the correlations seem weaker, and the insulation is again due to some form of magnetism; in this case, the spin-orbit coupling most likely does play an important role. But it should be less important in hetero-bilayer TMD material we are focusing on, because here magnetic order seems less probable, for a variety of reasons. Experiments (Ref. 9) show that in this regime the magneto-resistance is very weak, and the excitonic sensors do not detect local magnetic moments. Since spin-orbit coupling generally plays a most prominent role concerning various forms of magnetic order, our minimal model ignores it. That does not mean that it could not play some role, but in this work we aim to explain the main transport trends, which presumably do not require the inclusion of spin-orbit effects.

The possible role of spin-valley degrees of freedom within our proposed mechanism has not been investigated so far, however we expect that the main trends would remain robust even if it is included. This interesting line of investigation, in the context of other moiré materials/regimes, is left for future work.

R1.5

4. Regarding CPA, why does this local approximation work so well in reproducing the experimental results for this particular system in contrast to others?

Response R1.5

As a matter of fact, the entire field of metal-insulator transitions with disorder has not been studied experimentally in sufficient detail so far, and it is not yet completely clear what physical mechanism or theoretical picture should dominate the most robust features. While the non-interacting systems with disorder have been studied in great detail, the corresponding picture of Anderson localization does not seem to explain almost any available experiments. Recent work, however, suggested that adding certain interaction effects at a non-perturbative level can completely change the transition behavior in presence of disorder. This effect is due to a massive reorganization of the effective disordered potential, leading to a spectral gap opening at the Fermi energy, as seen in most experiments. This result was recently established within a theory that includes both localization and interaction effects, within the so-called TMT-DMFT formulation (Ref. 17), although the mechanism persists even at the simpler CPA level (Ref. 14), where localization is ignored. The general details of this new theory will be discussed elsewhere, but here we note that this is conceptually the simplest theory that non-perturbatively treats the interplay of interactions and disorder. This is the setup we utilize in this paper, which formally becomes exact in the limit of large coordination.

As any mean-field theory, it cannot be expected to accurately describe the immediate vicinity of the critical point, but presumably it should suffice over a broad parameter range surrounding the transition – precisely as in the current experiments. Systematic correction to this theory can be obtained by performing a certain loop expansion around an appropriate saddle-point corresponding to the mean field limit. Work along these lines is in progress, however the experimental test of such an “asymptotic” critical regime will likely require a much more refined generation of experiments, which are not available at the moment. Still, the capability to accurately describe what appears to be a mean-field regime around a disorder-dominated metal-insulator transition is a significant advance, and the current work is the first example of close agreement between theory and experiment in almost any system displaying disorder-dominated metal-insulator transitions. Motivated by the Reviewer’s question, we added a discussion about the possible limitations of our CPA-DMFT theory to the Supplementary Material (Line S35 - S41).

R1.6

5. Disorder can have different origins, such as charge impurities or twist angle disorder (although for aligned bilayers this is expected to be small). Can all sources of disorder be treated on the same footing by the distribution used here? Interactions introduce a self-energy that renormalizes the disorder, how is the disorder distribution renormalized as δ changes?

-- As a side comment, maybe the same kind of experiments on heterobilayers rotated at a small twist angle could reach the strongly disordered regime (larger values of λ) where the criticality is expected to be different. --

Response R1.6

Our work considers the situation with moderate disorder, a limit where its precise form and/or microscopic origin of disorder is generally expected to be of lesser importance. The rationale for doing so is the fact that the residual resistivity (further on the metallic side) is experimentally seen to be rather modest. Therefore, the bare amount of disorder is presumably small in these materials, even though its impact increases as the transition is approached. We agree with the Reviewer that one may envision situations where (bare) disorder is much stronger and where its form might play a more significant role, which is an interesting direction for the future work.

We should mention that although our CPA-DMFT setup can - in principle - describe different forms/types of disorder, its form generally suffers significant renormalizations close to the transition. Due to certain polaronic effects (Ref. 14) any smooth distribution of disorder ends up renormalizing to a bimodal (binary-like) form, facilitating the opening of a spectral gap at the Fermi energy at any filling. This effect, which is a robust feature, at least within our DMFT-type mean-field formulation, suggests that the results should be quite robust to the precise form of bare disorder. While this behavior was documented for certain models in recent work (Refs. 14 and 17), it is important to more carefully establish if different behaviors can still arise for certain special forms of disorder and/or interactions, especially if the disorder is strong. Work along these lines is currently in progress, but we do agree with the Reviewer that the corresponding experimental tests could also be enlightening. To clarify these important issues stressed by the Reviewer, we added an appropriate comment within the revised Supplementary Material (Line S42 – S48).

R1.7

6. What is the scaling for the calculated DOS? In some parts of the text it says $\rho \sim \omega^{1/3}$ and in some others $\rho \sim \omega^3$.

Response R1.7

We thank the Reviewer for pointing out these typos. The DOS critical exponent, within CPA+DMFT theory, is $\rho \sim \omega^{1/3}$. These typos have now been corrected.

R1.8

7. By looking at the very good agreement between theory and experiments, most probably it is not necessary to go beyond mean-field theory to describe the general trends of this transition. However, can the authors discuss in which limits/conditions should their approximation break down?

Response R1.8

Just like any other mean field theory, we expect CPA-DMFT we use to break down very close to the transition. Better understanding the corresponding “asymptotic” regime remains an interesting question for future work. The present setup can be obtained, as it turns out, as a saddle point approximation to a certain field theory, which in principle allows for systematic corrections to mean field theory. Work along these lines is in progress, although – as the Reviewer notes - the current setup should suffice to describe the current experiments. Following the Reviewer’s suggestions, we added a discussion of the limits/conditions for the validity of our theory to the Supplementary Material (Line S35 - S48).

R1.9

8. In the data from device 1 in Fig. S6, A does not become negative, is this attributed to a less disordered sample? Are estimates of the disorder amounts within each device available?

Response R1.9

We have strong reasons to believe that, for any continuous disorder-dominated metal-insulator transitions, the slope A should generically become negative before reaching the critical point from the metallic side. This is also found in all known experiments on similar transitions in other systems. In the data from Device 1, there exist only a few curves sufficiently close to the transition point, which does not provide detailed information. This is exactly the reason why we requested more experimental data from the Cornell group, and which they kindly provided (Device 2). These additional results, which were obtained on a different sample, do contain many more data points closer to the transition. The corresponding analysis indeed demonstrates that the coefficient A does become negative closer to the transition - in precise agreement with our theory.

As to the estimates of the disorder strength, one can look at the residual ($T=0$) resistivity in the metallic regime. The ratio of the high-T resistivity and the residual resistivity is typically used to quantify the amount of disorder in a given material, because in metals the high-T resistivity is dominated by thermal scattering (often due to phonons) and not by disorder, while the residual resistivity is controlled (mostly) by disorder. For Device 2, this ratio (using $R_{(T=50K)} / R_{(T=1.6K)}$) of the most metallic curve is around 5, while for Device 1 it is around 8, which is comparable. Since strong disorder corresponds to $R_{(high\ T)} \sim R_{(low\ T)}$, we conclude that both devices have a moderate amount of (bare) disorder, at a similar level. Motivated by the Reviewer’s question, we have added a discussion about this to the Supplementary Material (Line S84 – S92).

R1.10

9. I recommend adding the reference Phys. Rev. Lett. 121, 026402 next to reference 7 in the introduction, this was the theoretical prediction of Hubbard model physics in moiré TMDs. Reference 7 was the experimental confirmation.

Response R1.10

We thank the Reviewer for suggesting this important reference, which has now been added to the revised manuscript.

R1.11

10. In the caption of Fig. 1, could the authors mention what the vertical black and red lines stand for in panels B and E. It is said in the main text but not on the figure.

Response R1.11

We thank the Reviewer for the good suggestion. The explanation for these two lines has now been added to the caption of Figure 1.

R1.12

11. In the caption of Fig. S6, "three characteristic properties A, T and Δ ..." should be σ_o instead of A.

Response R1.12

We thank the Reviewer for pointing out this typo. A has now been corrected to σ_o .

R1.13

12. After Eq. (S5) it reads "hybridization" instead of hybridization. After Eq. (S10) "bonson" probability distribution.

Response R1.13

We thank the Reviewer for pointing out the typos. They have now been corrected.

Reviewer #2 (Remarks to the Author):

R2.1

In their manuscript "Disorder-dominated quantum criticality in moiré bilayers" the authors Yuting Tan, Pak Ki Henry Tsang, and Vladimir Dobrosavljevic propose and investigate a model for quantum criticality in moiré bilayers of transition metal dichalcogenides (TMDCs). In contrast to the strongly correlated regime, where the systems is dominated by the Mott-Hubbard physics, at the doping levels of interest in this work, the metal-to-insulator transition (MIT) is found to be disorder-driven. The results of the manuscript are interesting to a broad readership of condensed matter physicists and represent an important step towards the understanding the physics of moiré TMDcs. The text is clearly written and the scientific analysis of the model seems

to be sound. I therefore recommend the publication in Nature Communication if the following points are properly addressed:

Response R2.1

We thank the Reviewer for recommending publication, and the very helpful suggestions which led to the rather significant improvements of our manuscript.

R2.2

Main points

1. The central point of the paper is the explainability of quantum critical behavior at integer filling of moiré TMDCs by disorder effects. In the manuscript I miss a discussion of what is actually known about disorder in the (experimentally investigated) TMDC devices. This could include, for instance, the type of the disorder and the disorder strength. Is the uniform disorder proposed by the authors applicable to the manufactured devices from experimental evidences or is it justified rather a-posteriori?

Response R2.2

Given the appreciable values of the residual resistivity on the metallic side, especially as the transition is approached, disorder clearly plays an important role. Its precise microscopic form is not yet known, although the upcoming experimental investigations (e.g., using various scanning probes) should be very informative. Some preliminary reports of appreciable amounts of microscopic disorder have been obtained from very recent STM studies (e.g., by Abhay Pasupathy at Columbia), but nothing definitive is available at the moment.

We should stress, however, that within our CPA-DMFT (or even TMT-DMFT) picture, the form of disorder is massively and generically renormalized close to the transition. Due to certain polaronic effects (see Refs. 14 and 17) any smooth distribution of disorder ends up renormalizing to a bimodal (binary-like) form, facilitating the opening of a spectral gap at the Fermi energy at any filling. This effect, which is a robust feature, at least within our DMFT-type mean-field formulation, suggests that the results should be expected to be quite robust to the precise form of bare disorder. While this behavior was documented for certain models in recent work, it is important to more carefully establish if different behaviors can still arise for certain special forms of disorder and/or interactions, especially if disorder is very strong. Work along these lines is currently in progress, but we do agree with the Reviewer that the corresponding experimental tests could also be enlightening. To clarify these important issues stressed by the Reviewer, we added an appropriate comment within the revised Supplementary Material (Line S35 – S48).

R2.3

2. In the introduction the authors discuss nicely the differences of their modellization of quantum criticality via disorder with the Mott criticality. The later was recently investigated in detail

within a dynamical mean-field theory study of the so-called moiré Hubbard model in a preprint of Jiawei Zang, Jie Wang, Jennifer Cano, Antoine Georges, and Andrew J. Millis (arXiv:2112.03080). The authors should contrast their results to the ones obtained in this preprint. Especially, there, the authors also demonstrate scaling behavior in both the metallic and insulating part of the phase diagram. Please also comment on the “mirror symmetry” in this respect.

Response R2.3

We thank the Reviewer for suggesting this important reference, which we now added to the paper. We now add an additional section about the “Degree of mirror symmetry of the scaling function” in the Supplementary Materials (Line S122 -S146), to further illustrate and explain this concept. In this section, we introduce a notion of MSI (Mirror Symmetry Index), which can serve as an indication of the “degree of mirror symmetry”, reflecting the relative significance of strong correlation effects. In our case, MSI is just 1.6, which is much smaller than that in the Mott case, which is discussed in the reference pointed out by the Reviewer (whereas in previous studies of the Mott regime $MSI > 10$). This is not unexpected because Mott physics plays a significant role at filling $f=1$, but not around $f=2$ (near a band insulator), which we consider in this work. The smallness of the MSI in the disordered case is not surprising since, in presence of disorder, the temperature dependence of the resistivity on the metallic side is generically much weaker than on the insulating side. This is in dramatic contrast to a very notable resistivity drop on the metallic side, in the Mott regime. Physically, it reflects the thermal formation/destruction of heavy quasiparticles (Ref. 35), an effect well established both within the DMFT-type theories, and by various experiments in the Mott regime. Consequently, MSI is much larger in the Mott regime, as one of the authors theoretically predicted in 2011 (Ref. 25). This prediction was later confirmed by a beautiful series of experiments by K. Kanoda in Mott organics (Ref. 26). The new data provided in Ref. 9 on TMD moiré hetero-bilayers are fully consistent with this early prediction in the Mott regime ($f = 1$), but not in the disorder-dominated regime ($f = 2$) we consider here.

Another notable difference between the two situations is that the proposed (and observed!) scaling behavior in the Mott regime exists only at temperatures $T > T_c$, i.e. **above** the metal-insulator phase coexistence region. This phenomenology is well established within the DMFT theories (Ref. 26), but it has also been experimentally observed in various Mott systems, for example in Mott organics materials. It is worth noting, however, that T_c is typically very small, on the scale of a percent of the electronic bandwidth. Since in moiré materials the bandwidth is on the scale of 100-200K, if a phase coexistence region existed there, it would be confined to $T < T_c \sim 1-2K$, which is at the boundary of the current reach of the experiments. In contrast, in the case of disorder-dominated metal-insulator transition, one does not expect to have any phased coexistence region, since the latter is known to be quite vulnerable to disorder, especially in $d=2$ (based on the well-known Imry-Ma arguments). In the disordered case we consider here, we therefore expect the scaling to extend all the way down to $T=0$, as we explicitly show within our CPA-DMFT solution, and consistent with the experiments.

R2.4

3. In Fig. 4 the authors show a nice phase diagram as a function of temperature and band splitting. I was wondering how this phase diagram would be changing if the disorder strength and/or the electron-boson coupling were changed? Is the existence of the Mooji region bound to a specific choice of the (relative) values of these parameters? In this respect I was a bit confused by the notion of “comparison to experimental findings without any adjustable parameters” (l. 156-157). The authors should clarify this statement.

Response R2.4

Within our model and the current theory setup, we are certain that changing these parameters will only result in quantitative, but no qualitative changes in the phase diagram. Certain features of the corresponding critical regime indeed display a significant degree of universality, in agreement with general expectations for critical phenomena. To be more precise, when performing the scaling analysis, we are plotting the phase diagram in terms of dimensionless quantities: the dimensionless distance to the transition, the reduced resistivity R/R_c , etc. In this way we reveal the form of the scaling function, which is a universal feature characterizing the critical regime, similar to the (universal) values of the critical exponents. This we believe is the reason why the scaling function extracted from experiments does quantitatively agree with our model, without adjusting any parameters (we did not tune the model parameters to match the experiment!). Indeed, the same theoretical scaling function agrees precisely with data from both experimental samples, although no two samples are precisely the same (e.g., in terms of the precise level of disorder, etc.).

Concerning the Mooij point, it is generally believed to be a generic feature of all disorder-dominated transitions. Indeed, in all known (experimental) examples of disorder-dominated transitions, it is always found that the Mooij point always arises not too far from the critical point. Within our model, the emergence of a Mooij point it is also a generic/robust feature, although its precise location (relative to the transition itself) may be parameter dependent.

R2.5

Minor points (and typos, etc.)

4. *What is the filling f referring to? Is this the filling with or without displacement field?*

Response R2.5

We are following here the notation used in the paper by Shan and Mak, in which the filling f indicates the number of electrons per moiré cell. So the Mott insulator corresponds to $f = 1$, and the band insulator corresponds to $f = 2$. In this work we are focusing on the metal-insulator transition at fixed filling $f = 2$, while the displacement field is varied in such a way that only the bandwidth is changing, as demonstrated in Ref. 9.

R2.6

5. *Is there any sign of a magnetic ordering tendency appearing in the CPA-DMFT calculations?*

Response R2.6

For simplicity, we have neglected the magnetic (spin) degrees of freedom in our model. We have done so because there exists no experimental evidence of any kind of magnetism in the $f = 2$ regime, nor do we expect it to be significant here. There are two experimental features that suggest this picture: (a) The magnetoresistance is very weak here (Ref. 9), in dramatic contrast to Mott regime ($f = 1$); (b) The recently developed excitonic sensor technique was also used in Ref. 9, to very convincingly detect local magnetic moment formation (Curie's Law for the spin susceptibility) at $f = 1$, but not around $f = 2$. Consequently, we do not expect magnetism to play any important role in the regime we investigate, nor can it arise within our setup. We added a comment in the manuscript to make this point clearer (Line 86).

R2.7

6. *Typos*

- l. 12: “on in” → “on”.
- l. 26: “TMD” → “transition metal dichalcogenide (TMD)”.
- l. 35: “here indeed proved” → “has indeed proven”.
- l. 127: “the the” → “the”.
- l. 221: “examples strongly” → “examples of strongly”.
- l. S11: “bosons) for triangular” → “bosons) for the triangular”.
- l. S19: “hybridazation” → “bosons) for the hybridization”.
- l. S25: “green” → “Green”.
- l. S74: $R_c \rightarrow R_c$.
- l. S76: “give” → “gives”.
- l. S88: “mention” → “mentioned”.
- Fig. S2: “Color” → “Colors”.

Response R2.7

We thank the Reviewer for pointing out the typos, which all have been corrected.

List of changes:

1. Line 83 - 86, 222: we modified the discussion about low energy bosons.
2. Line 207 - 216: we added a discussion about neglecting the on-site Coulomb repulsion U , as well as the long-range component of the Coulomb interaction.
3. Line S35 - S48: we added a discussion of the limits/conditions for the validity of our theory to the Supplementary Materials.
4. Line S84 – S92: we added a discussion concerning the levels of disorder in the experimental Device 1 and Device 2.
5. Line S122 - S146: we added a new section entitled “Degree of mirror symmetry of the scaling function” to the Supplementary Material.
6. New figure: Figure S11.
7. Fig.1: we added an explanation for the dashed lines in the caption for Figure 1.B and 1.E.
8. We added new references 7, 21, 22, 28, 29, 31, 34, and 35 to the manuscript.
9. Typos have been corrected.

REVIEWERS' COMMENTS

Reviewer #1 (Remarks to the Author):

I thank the authors for the detailed explanations that they have provided to my questions and I believe the modifications they made to the manuscript have improved its quality.

It seems to me that line 34 still contains a typo -- The word here appears two times

"Here, the magnetic field response here has indeed proven to be remarkably mild, suggesting the lack of ... "

That being said, I recommend the publication of the article in Nature Communications.

Reviewer #2 (Remarks to the Author):

In the revised version of their manuscript "Disorder-dominated quantum criticality in moire{'e} bilayers" and in the corresponding reply to the referees, the authors Yuting Tan, Pak Ki Henry Tsang, and Vladimir Dobrosavljevi{'c} addressed all the points I have previously raised. Hence, I now can recommend the publication of the manuscript in Nature Communications.